

# Ground motion emissions due to wind turbines: observations, acoustic coupling, and attenuation relationships

Laura Gaßner[1] and Joachim Ritter[1]

[1]Karlsruhe Institute of Technology, Geophysical Institute, Hertzstraße 16, 76187 Karlsruhe, Germany

**Correspondence:** Laura Gaßner (laura.gassner@kit.edu)

**Abstract.** Emissions from wind turbines (WT) cover a wide range of infrasound and ground motions. Some of these emissions are perceived by local residents and can become a source of disturbance or annoyance. To mitigate such disturbances, it is necessary to better understand and, if possible, suppress WT-induced emissions. Within the project Inter-Wind we record and analyze ground motion signals in the vicinity of two wind farms on the Swabian Alb in Southern Germany, simultaneously
with acoustic and meteorological measurements, as well as psychological surveys done by cooperating research groups. The investigated wind farms consist of three and sixteen WTs, respectively, and are located on the Alb peneplain at 700-800 m height, approximately 300 m higher than the municipalities. The main aim is to better understand reasons why residents may be affected from WT immissions, based on interdisciplinary data, methods, and expertise.

Known ground motions include vibrations due to eigen modes of the WT tower and blades, and the interaction between the
passing blade and the tower, causing signals at constant frequencies below 12 Hz. In addition, we observe signals in ground motion recordings at frequencies up to 90 Hz which are proportional to the blade-passing frequency. We can correlate these signals with acoustic recordings and estimate sound pressure to ground motion coupling transfer coefficients of 3-16.5. Sources for these emissions are the WT generator and possibly the gearing box. The identification of such noise sources can help to design potential counter-measures in order to increase the public acceptance of WTs. The measurements in the municipalities
indicate that WTs are perceived more in the location where the wind farm is closer to the municipality (approx. 1 km). However, there is also a major railway line which produces higher vibration and infrasound signal amplitudes compared to the WTs.

Along the measurement lines the decay rate of the WT-induced ground motions is determined for a damping relation proportional to $1/r^b$. We find frequency-dependent $b$-values for different scenarios at our geological setting of Jurassic limestone on marl, sandstone, and Quaternary deposits. These damping relationships can be used to estimated emissions in the far-field
and to plan mitigation strategies.

## 1   Introduction

To achieve a transition to renewable energies an increased utilization of wind energy is necessary. In Germany the number of newly installed wind turbines (WT) increased to more than two thousand (>1 MW) per year in 2017, but has since dropped to roughly 20 % of that number in the last four years (Bundesnetzagentur, 2020). The drop in expansion can be related to more
restrictions, e.g., in Bavaria the distance to dwellings must be 10 times the height of the WT, or protests of conservationists



and residents. Opponents protest against environmental, visual, and acoustic aspects of WT installation and operation due to a feared loss of their quality of life. Therefore, due to lawsuits many WT projects are delayed significantly (FA Wind, 2019). WT emissions may also influence technical operations such as high-resolution electron microscopes, air traffic control or seismological observatories. Therefore, we would like to derive general properties of WT emissions based on dedicated
highly-sensitive field experiments.

Within the project Inter-Wind we aim to classify which properties of WT operation could be responsible for the annoyance of residents in the vicinity of two wind farms on the Swabian Alb, a mountain range in SW Germany (compare Figure 1). Beside psychological questionnaires we measure meteorological, acoustic, and ground motion data which build the foundation for an interdisciplinary analysis (Gaßner et al., 2022). In this paper we concentrate on the seismological perspective, focusing
on induced ground motions of WTs. These are caused by the coupling of the WT foundations with the ground (Nagel et al., 2021) and are expected not to be perceptible by humans (e.g., Ratzel et al., 2016). Regulations for the impact of vibrations on humans in buildings are defined in DIN 4150-2 (1999), stating a relevant frequency range of 5.6 Hz to 80 Hz. As a lower limit for perceptible signals, typically a value of 0.1 mm/s (=100 $\mu$m/s) is taken, derived from specifications in DIN 4150-2, table 1. Although these amplitudes are unlikely to be reached, surveys of residents living near wind farms have shown that vibrations
are named as an annoying property of WT operation (e.g., Michaud et al., 2016) and measured data is needed to demonstrate the opposite.

Signals emitted by WTs are related to the eigen modes of the WT tower and blades (Nagel et al., 2019; Zieger et al., 2020), and the blade-passing frequency (BPF) and its multiples (e.g., Styles et al., 2005; Nagel et al., 2019; Neuffer et al., 2021). While the latter are proportional in frequency to the rotation rate of the WT, natural vibrations due to eigen modes are of
constant frequency and increase in amplitude with higher rotation rates. Typically the emitted ground motions were studied for frequencies below 20 Hz. Lower frequencies are attenuated less than higher frequencies and can be detected up to distances of more than 10 km (e.g., Schofield, 2001; Saccorotti et al., 2011). Seismic signals caused by WT operation are studied eagerly in the seismological community, as they disturb sensitive measurements, e.g., monitoring networks (Styles et al., 2005; Stammler and Ceranna, 2016; Estrella et al., 2017; Neuffer and Kremers, 2017; Zieger and Ritter, 2018).
Our ground motion measurements were conducted within two campaigns aimed at characterizing WT signals in the vicinity of two wind farms. They were set up in combination with acoustic measurements to provide a data basis to evaluate annoyance reports of local residents (Gaßner et al., 2022). The wind farms under investigation are located on the Swabian Alb, near Geislingen an der Steige (compare Figure 1), and consist of three and sixteen WTs of the same type, respectively. The WT type is a General Electrics GE 2.75-120 with 139 m hub height, 120 m rotor diameter, and 2.78 MW rated power.

**2  Measurements**

We installed seven to ten seismic recording stations in profile-like setups (Figure 1). For comparability, we chose profiles of southwest-northeast orientation to achieve an observational connection between the place of emission (near the WTs) and immission (at local residents) at both wind farms. Both profiles are located across significant changes in elevation, the so-



called Alb Cuesta: the WTs are placed on the Alb peneplain at approx. 700 m to 800 m elevation on Kimmeridgian limestone
(Upper Jurassic), whereas the villages of Kuchen and Degenfeld are located in the valley at ca. 400 m on Quaternary deposits
overlying marl and sandstones (Middle Jurassic) and at ca. 500 m on Quaternary deposits overlying marl (Lower Jurassic),
respectively. At wind farm Tegelberg with three WTs, a total recording time of 3.5 months was achieved with five field stations
and additional four 2-week long measurements at the residents in the municipality of Kuchen. At wind farm Lauterstein four
to six weeks of measurements were conducted, with a total of seven stations (five field stations and two in Degenfeld).

Close contact with the WT operators was established to support measurements and signal analysis, and to provide WT data.
This WT operating data was averaged over 10 minute time windows. For each campaign one recording station was installed
inside the tower of one WT. The other stations were placed in a profile-like setup towards the nearest settlement, i.e., where
complaints were registered. Within these communities residents had been surveyed via psychological questionnaires earlier to
find participants offering their houses for further measurements (Gaßner et al., 2022).

Details on the instrumentation of each recording station are summarized in Table 1. CUBE[3] data loggers were utilized for all
field stations where there was no power available. These recording stations were mostly buried in the ground in approximately
30 cm depth for insulation (except IW03G which was installed inside a cellar). For the campaign in Tegelberg recorders were
set to a sampling rate of 100 Hz, in Lauterstein 200 Hz was used.

**Table 1.** Sensors (Streckeisen STS-2, Nanometrics Trillium Compact Posthole (TC-PH), Lennartz LE-3D) with their respective eigen period
and data loggers (Earth Data PR6-24 Portable Field Recorder (EDL), Digos DATA-CUBE[3], Nanometrics Centaur Digital Recorder) used for
the measurement campaigns at Tegelberg (IW02) and Lauterstein (IW03).

| | Sensor | | | Datalogger | | |
|---|---|---|---|---|---|---|
| | STS-2 120 s | TC-PH 20 s | LE-3D 1s | EDL | CUBE | Centaur |
| IW02A | x | - | - | x | - | - |
| IW02B, C | - | - | x | - | x | - |
| IW02D, E | - | x | - | - | x | - |
| IW02F, G, H, I, J | - | x | - | - | - | x |
| IW03A, F | - | x | - | - | - | x |
| IW03B, C, D | - | x | - | - | x | - |
| IW03E, G | - | - | x | - | x | - |

## 2.1 Wind farm Tegelberg

The first measurement campaign at wind farm Tegelberg took place from 2020-10-20 to 2021-02-05, amounting to 108 days.
In total we deployed ten recording stations, of which one station (IW02A) was installed in the tower of WT 1, one station
(IW02B) near an acoustic measurement site in approximately 150 m distance to WT 1, and three more stations in the forest
surrounding the wind farm (IW02C-E, at 150 m to 550 m distance to the nearest WT). Additionally, we conducted two-
week long measurements outside four resident houses (IW02F-J) and in the basement of resident 3 (IW02H) in Kuchen. The





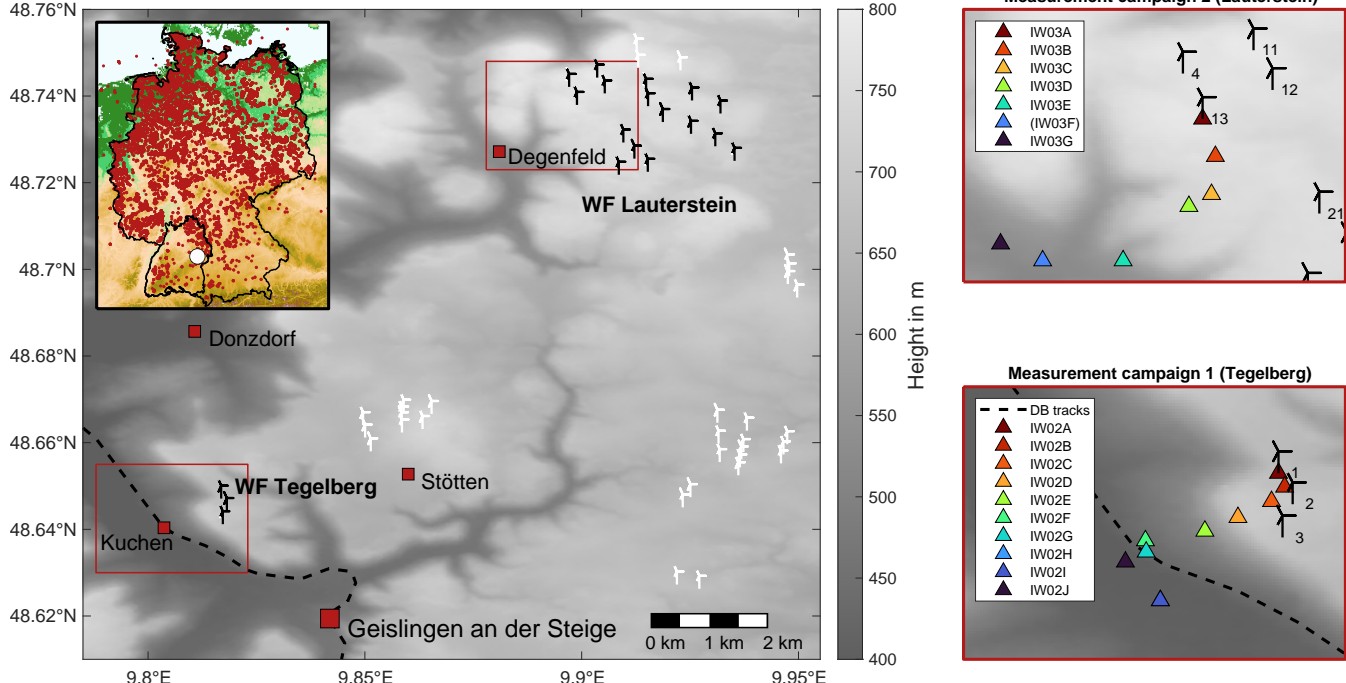

**Figure 1.** Inter-Wind study area with relevant places and locations of WTs (black - part of this study, white - not studied here). Inset: Map of Germany with the outline of the state of Baden-Württemberg. The white marker denotes the location of the project area, dark red dots mark German WT positions (Bundesnetzagentur, 2020). Detailed maps show the recording station positions (colored triangles) for measurement campaigns at wind farms Tegelberg (bottom) and Lauterstein (top). The dashed line (DB tracks) is the main railway line Stuttgart-Ulm.

distribution of the recording stations towards the southwest, as seen from the wind farm, is shown in Figure 1. Wind farm Tegelberg belongs to the town of Donzdorf and is located approximately 4.5 km to the south of it. The average distance of the affected residents in Kuchen to the wind farm is 1 km. Between the inhabited area in the valley and the wind farm Tegelberg there is a difference in topography of approximately 300 m.

    Figure 2a gives an overview of the relation between rotation rates and wind speed at WT 1 of wind farm Tegelberg. Fur-
thermore, the wind direction distribution is shown, exhibiting a dominance of westerly wind conditions. Rotation rates for the considered turbine type typically reach up to 12.5-12.6 rotations per minute (rpm) during full operation at wind speeds above 6 m/s, which corresponds to 0.625-0.630 Hz primary BPF. Additionally, a plateau at 7.9-8 rpm can be observed, corresponding to a BPF of 0.395-0.400 Hz, related to wind speeds between 2 m/s and 6 m/s. For the other two WTs at Tegelberg similar observations can be made. During night time (21:00 to 5:00 UTC, 22:00 to 6:00 local time) WT 3 runs in a noise-reduced
mode to mitigate acoustic emissions, resulting in a maximum of 12 rpm at full load (0.6 Hz BPF).



## 2.2 Wind farm Lauterstein

At wind farm Lauterstein measurements were conducted for 46 days, from 2021-02-22 to 2021-04-08. We deployed seven recording stations, though at one station only very limited data could be recovered due to problems with the sensor (IW03F). Again, one recording station was installed inside the tower of WT 13 (IW03A, compare Figure 1), and three stations in the surrounding forest (IW03B-D), in distances of 400 m to 900 m to WT 13. Additionally, one station was set up near the village Degenfeld (IW03E, 1.5 km to WT 13) and one station inside a community building (IW03G, 1.9 km to WT 13). As no significant complaints were reported with regard to emissions from wind farm Lauterstein, no affected residents took part in the campaign.

A noise-reduced mode is implemented at WT 13, due to a dwelling in 550 m distance. This leads to rotation rates limited to 11 rpm (0.55 Hz BPF) during night time (Figure 2b). Two more WTs (WT 12 and WT 21, compare Figure 1) close to WT 13 also have noise-reduced operation, with maximum rotation rates at 11.5 rpm (0.575 Hz BPF) at night. Wind direction distribution at Lauterstein (Figure 2 b) is similar to wind farm Tegelberg, with west to westnorthwest the dominant direction.

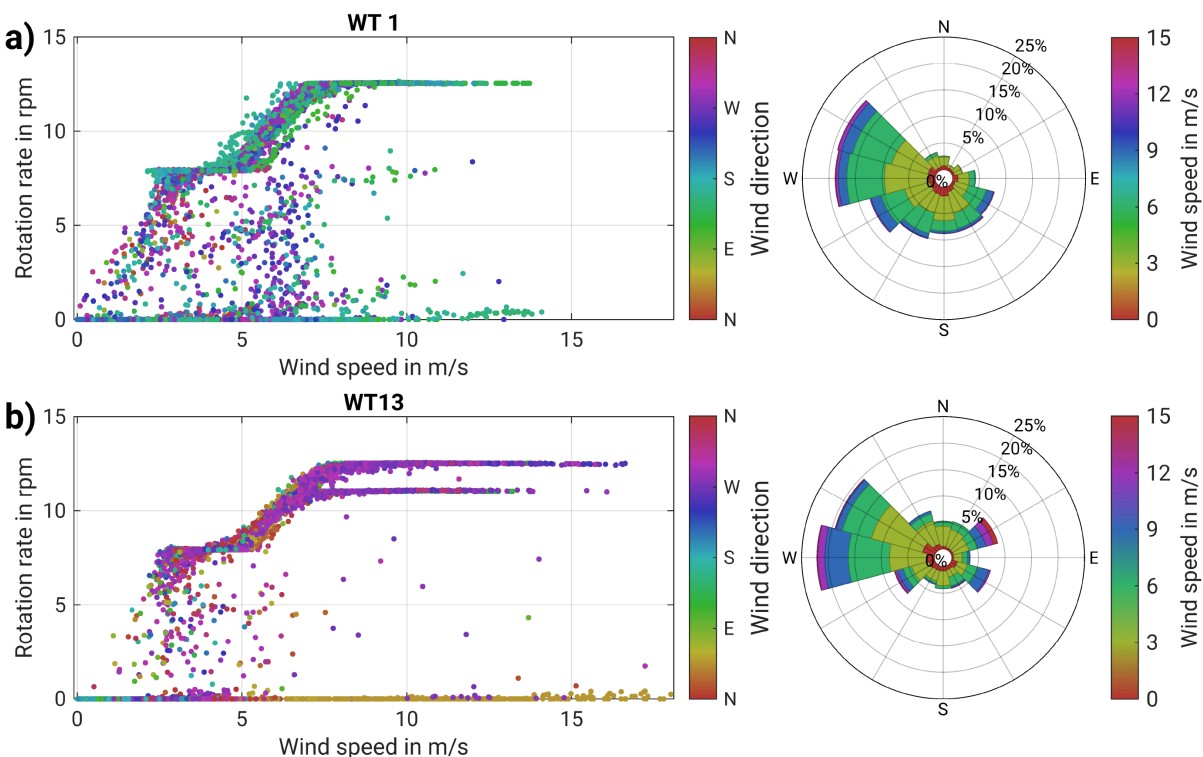

**Figure 2.** Operating data for a) WT 1 of wind farm Tegelberg, and b) WT 13 of wind farm Lauterstein. Left: rotation rate vs. wind speed, color-coded by the wind direction. Right: wind rose for the wind speed and direction distribution during the measurement period. The outer circle represents 25 % occurrence, indicating that W and WNW were the main wind directions with approx. 19 % (Tegelberg) and 22 % and 19 % (Lauterstein) likelihood each.





## 3    Ground motion signals

From the raw data recovered with our recorders, the true ground motion is calculated by removing the instrument response
using the *ObsPy* (Krischer et al., 2015) package and the respective instrument information. A 2-pole bandpass filter is then
applied with corner frequencies of 0.1 Hz and 45 Hz or 95 Hz, for data from Tegelberg and Lauterstein, respectively. We
calculate power spectral densities (PSD) based on the method of Welch (1967), taking 60 s time windows with an overlap of
20 s.

Typical signals measured in the vicinity of wind farms are the eigen modes of the WT tower and blades (Nagel et al., 2019),
which are of constant frequency and exhibit amplitudes proportional to the rotation rate of the WT. The eigen modes occur
mainly at frequencies below 12 Hz and, as they are less damped than higher frequencies, are therefore detectable over long
distances (Saccorotti et al., 2011). Such signals lead to a potential reduction of detection capabilities of monitoring networks,
e.g. for microseismicity (Neuffer and Kremers, 2017). As we study two wind farms consisting of the same turbine type, we
expect to find the same frequencies in both measurements.

In Figure 3 we present an overview of ground motion signals recorded during a three hour time window at wind farm
Tegelberg (20:00 - 23:00 UTC, 21:00 - 24:00 local time). It includes a drop in rotation rate of all three WTs, with a time
interval when all WTs shut down and then restart after approx. 15 min (Figure 3a). The reduction of rotation rate can be
explained by a drop in wind speed from approx. 8 m/s to 5 m/s. It is not obvious why the WTs turned off completely for a short
time, though. Typically, the detection of protected species, e.g., bats or red kites, in the vicinity of the WTs can be a reason
for sudden automated shutdowns. The change in rotation rate is visible in the ground motion data (Figure 3b), and as expected
the change in amplitude is less pronounced with increasing distance of the recording stations to the WTs. At 360 m distance
ground motion signal amplitudes are approximately 1 $\mu$m/s during WT operation.

In the spectrograms the signals and their relation to frequency can be observed well, for the station inside WT 1 (IW02A,
Figure 3c) and a site in approx. 360 m distance to the next WT (IW02D, Figure 3d). At IW02D signals proportional in frequency
to the BPF are visible for each WT between 20:30 UTC and 22:00 UTC when the rotation rate fluctuates between 12.5 rpm and
8 rpm. One of the observed signals corresponds to the generator speed with frequencies at $46\times$BPF (II, Table 2), other signals
are at $32\times$ (I), and $64\times$BPF (III). They are most prominent for the two main rotation rates in the operating range, at 8 rpm and
12.5 rpm. A more detailed study of these signals can be found in Section 3.1 for wind farm Tegelberg.

Additionally, in Figure 3d signals are visible that only last for a few minutes and occur at frequencies below 10 Hz. They are
related to the train traffic in the valley where Kuchen is located, and are especially prominent at the resident locations (compare
IW02F in Figure 3b). At this location the train signals reach ground motion amplitudes of more than 20 $\mu$m/s. A more detailed
discussion of these signals can be found in section 3.2.

At wind farm Lauterstein we have a wider spread of recording sites as well as WT distribution. Here, during a drop in wind
speed (compare Figure 4a), which leads to a reduction of rotation rates, we observe a visible reduction in ground motion signal
amplitude for the station inside WT 13 (Figure 4b). At the recording stations outside the WT there is only a very small reduction
in ground motion amplitude. The spectrograms (Figure 4c and d) indicate that signal amplitudes related to the BPF are low





outside the WT. Here, additional signals at $96\times$ (IV), and $128\times$BPF (V) can be observed, due to the higher sampling rate of 200 Hz at wind farm Lauterstein compared to 100 Hz for measurements at wind farm Tegelberg. The respective frequencies for these signals are listed in Table 2.

Figure 5 shows a direct comparison of ground motion spectra inside the WT towers (IW02A in WT 1 and IW03A in WT 13 at wind farms Tegelberg and Lauterstein, respectively) and in approx. 360 m and 390 m distance (IW02D and IW03B). Data of 2-3 days with full operation of the whole wind farms was considered for the calculation of the spectra. Both, the spectra of ground motion inside the WT tower and in the free field at less than 400 m distance, agree considerably well. The highest similarity can be observed for frequencies below 5 Hz (enlarged box in Figure 5), while at higher frequencies the spectra for

IW02D and IW03B differ by 5-25 dB. This difference can be explained by the slightly smaller distance (360 m compared to 390 m) to the respective WT for IW02D and the increased background signal level due to the proximity to Kuchen and the train tracks. A difference of up to 17 dB is visible for IW02A and IW03A at frequencies higher than 22 Hz.

**Table 2.** Frequencies corresponding to multiples of the BPF for the primary rotation rates of 7.9-8 rpm and 12.5-12.6 rpm and labels I-V used in Figure 3 and 4. For simplicity values are provided for 8 rpm and 12.5 rpm only.

| | 8 rpm | 12.5 rpm | label | | 8 rpm | 12.5 rpm |
|---|---|---|---|---|---|---|
| $1\times$BPF | 0.4 Hz | 0.625 Hz | I | $32\times$BPF | 12.8 Hz | 20.000 Hz |
| $2\times$BPF | 0.8 Hz | 1.250 Hz | II | $46\times$BPF | 18.4 Hz | 28.750 Hz |
| $3\times$BPF | 1.2 Hz | 1.875 Hz | III | $64\times$BPF | 25.6 Hz | 40.000 Hz |
| $4\times$BPF | 1.6 Hz | 2.500 Hz | IIIa | $65\times$BPF | 26.0 Hz | 40.625 Hz |
| $5\times$BPF | 2.0 Hz | 3.125 Hz | IV | $96\times$BPF | 38.4 Hz | 60.000 Hz |
| $6\times$BPF | 2.4 Hz | 3.750 Hz | V | $128\times$BPF | 51.2 Hz | 80.000 Hz |
| $7\times$BPF | 2.8 Hz | 4.375 Hz | | | | |
| $8\times$BPF | 3.2 Hz | 5.000 Hz | | | | |

## 3.1 Signals at place of emission

Amplitudes of the WT induced signals are generally proportional to the rotation rate, which in turn is related to the wind speed

as shown in Figure 2. Exemplary, we show spectra for recording station IW02B (150 m from WT 1 at wind farm Tegelberg) for two weeks of data in Figure 6, for rotation rates between 8 rpm and 13 rpm. Below 12 Hz we observe a steady increase of PSD values with rotation rate. Between 12 Hz and 20 Hz PSD values do not increase for higher rotation rates, but exhibit peaks at different frequencies for different rotation rates. A significant peak can be observed at 28.9 Hz and further increased PSD values between 33 Hz and 42 Hz. These spectral values correspond to signals with frequencies proportional to the BPF (above

12 Hz). As mentioned before, signals occur at $46\times$BPF (generator speed) and at multiples of $32\times$BPF (compare Table 2). In Figure 6 we find that the peak at $64\times$BPF is slightly smaller than a peak at $65\times$BPF which is considered for the choice of frequencies in the following evaluation. The peak at 28.9 Hz is visible at all rotation rates because for the sorting of the spectra







**Figure 3.** a) Operating data (rotation rate and wind speed) of the three WTs at wind farm Tegelberg for 2020-10-30. The black-dashed box indicates the time window chosen for b)-d). b) 3 h time series of the vertical ground motion for stations IW02A, IW02B, IW02D, and IW02F, with the respective distance to the nearest WT. Note the different amplitude scales. c) PSD spectrogram for the vertical ground motion of IW02A inside the tower of WT 1 for the same 3 h time window as b). d) same as c) for recording station IW02D, located in approximately 360 m distance to WT 3.

only the rotation rate of WT 1 is considered. Signals from the other WTs, potentially operating at different rotation rates, can be present. Furthermore, it has to be noted that rotation rates are averaged over 10 minute time windows and, therefore, can

span a larger rotation rate range.




**Figure 4.** a) Operating data (rotation rate and wind speed) of WT 13 at wind farm Lauterstein for 2021-02-25. The red-dashed box indicates the time window chosen for b)-d). b) 6 h time series of the vertical ground motion for recording stations IW03A, IW03B, IW03C, and IW03D, with the respective distance to the nearest WT. c) PSD spectrogram for the vertical ground motion of IW03A inside the tower of WT 13 for the same 6 h time window as b). d) same as c) for station IW03D, located in approximately 390 m distance to WT 13.



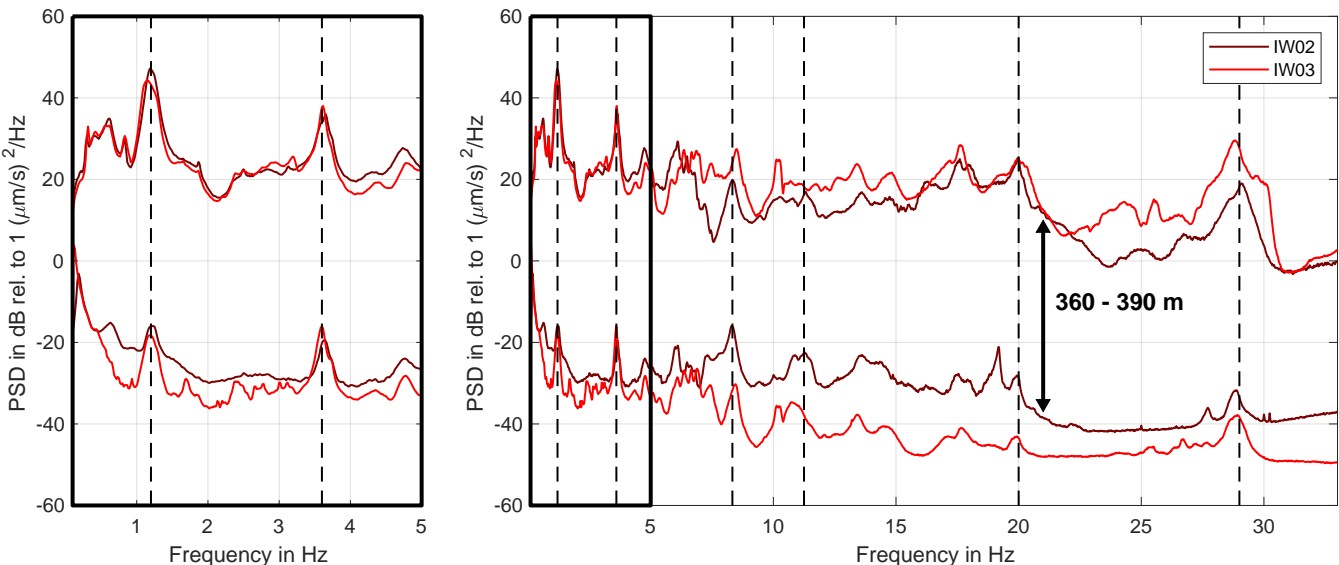

**Figure 5.** Comparison of mean PSD spectra for recordings inside the WT tower (upper curves) and close by in the free field (lower curves) for measurement campaigns Tegelberg (IW02, dark red, distance 360 m) and Lauterstein (IW03, red, distance 390 m). For Tegelberg all three WTs were in full operation for approximately two days. At Lauterstein all 16 WTs were in full operation for almost three days. Frequencies considered for amplitude decay analysis are marked by black-dashed lines.

To study further dependencies of signal amplitudes, we choose 25 two-hour- to 17-hour-long segments from 45 days (2020-10-29 to 2020-12-12) with rotation rates either constant at 8 rpm (549 ten-minute segments) or 12.5 rpm (366 ten-minute segments) and analyze the maximum vertical ground motion velocity ($v_Z$) values for three dominant frequencies per rotation rate. These frequencies are 12.5 Hz, 18 Hz, and 26 Hz for 8 rpm, as well as 20 Hz, 29 Hz, and 41 Hz for 12.5 rpm. Amplitude
values are taken from data filtered in frequency bands of $\pm 1$ Hz relative to the dominant frequencies.

We observe a distribution of amplitudes dependent on different wind direction segments (Figure 7). For rotation rates of 8 rpm, there are four distinct angular segments (E, S, W, and NW), while for 12.5 rpm there are only two zones (SE, and W), which implies that winds strong enough to achieve full WT operation originate either in western or south-eastern direction. The amplitudes of the generator frequency signals correspond to the first or second multiple of 32×BPF, 12.5 Hz (32×BPF at
8 rpm) and 41 Hz (64×BPF at 12.5 rpm), respectively. The WT is located at $-17°$ relative to North as seen from IW02B.

## 3.2 Signals at place of immission

As the main focus of our research project lies on how local residents experience WT emissions, we recorded two weeks of data at four different locations within the municipality of Kuchen. The measured ground motion signals, though, are dominated by the train traffic, with trains passing through the valley in intervals of several minutes, even at night times (compare Figure 3
b, IW02F). Due to the high frequency of train passages, an assignment to WT operating conditions with data available for ten




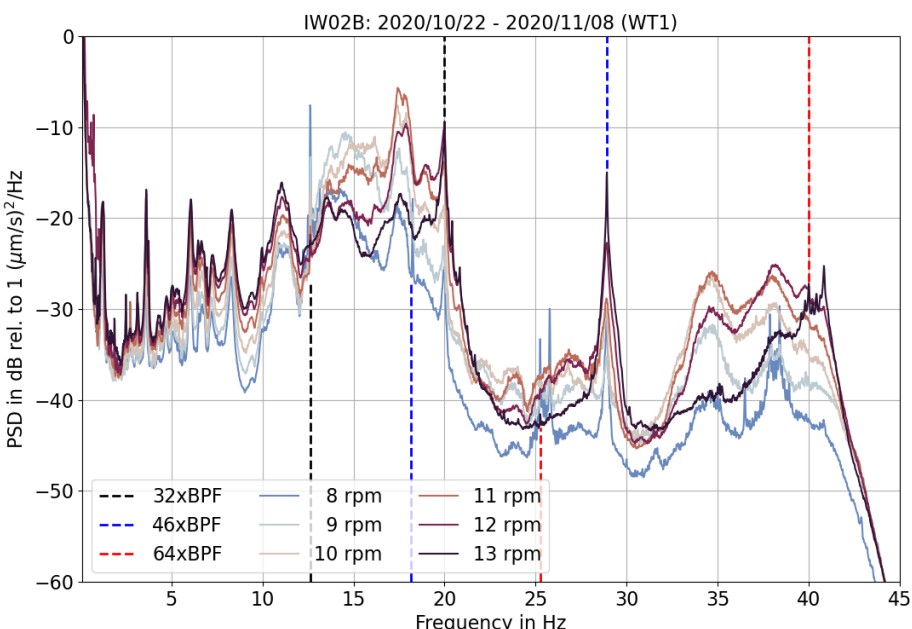

**Figure 6.** Mean spectra for recording station IW02B (150 m distance to WT 1 of wind farm Tegelberg) for a two week period, sorted by the rotation rate of WT 1. Only spectra for 8 rpm to 13 rpm are shown. Black, blue and red dashed lines mark peaks related to multiples of the BPF for 8 rpm (lines originating at the bottom) and 13 rpm (lines originating at the top), respectively (see Table 2). These frequencies are prominent in the ground motion data and their amplitude distribution relative to wind direction is shown in Figure 7.

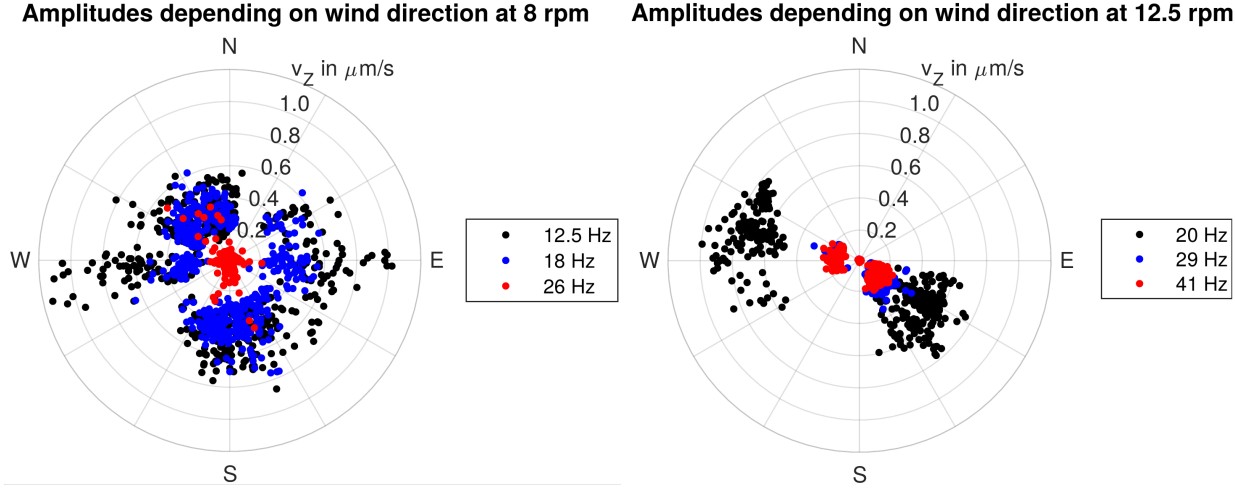

**Figure 7.** Maximum vertical ground motion amplitudes for 10 min time intervals for the frequencies marked in Figure 6 for station IW02B (150 m distance to WT 1 of wind farm Tegelberg), depending on wind direction, for 8 rpm (left) and 12.5 rpm (right).




minute intervals is therefore not meaningful. Each train passage excites a distinct signal, with amplitudes, signal duration and time-dependent shape related to the specific type of train (local, long distance and freight trains) as well as the direction of travel. The distance of the measurement positions relative to the train tracks ranges from 20 m to 300 m.

The overall amplitude distribution for each of the five recording stations (at one resident two instruments were installed) is shown in Figure 8. There is a noticeable difference to the spectra close to the WT (e.g., Figure 6), as signals related to the eigen modes of the WTs or the BPF can not be identified. Instead, a clear trend in increased amplitudes can be observed, related to the distance of the measurement position to the train tracks. The closest recording station (IW02F, 20 m distance) reaches PSD values of almost -120 dB (relative to 1 (m/s)$^2$/Hz, corresponding to 0 dB rel. to 1 ($\mu$m/s)$^2$/Hz) with the maximum at approx. 14 Hz. A decrease in amplitude and peak frequency is found with increasing distance for the frequency range 5-30 Hz.

At 300 m distance the maximum is observed at 9 Hz with a PSD value of -140 dB (relative to 1 (m/s)$^2$/Hz, corresponding to -20 dB rel. to 1 ($\mu$m/s)$^2$/Hz). Differences in the spectra for the indoor and outdoor instruments at resident 3 arise for frequencies above 25 Hz, with lower amplitudes of up to -10 dB indoors. Sharp peaks at approx. 20 Hz, as observed at residents 3 and 4, could be caused by household devices, e.g., a washing machine operating at 1200 rotations per minute.

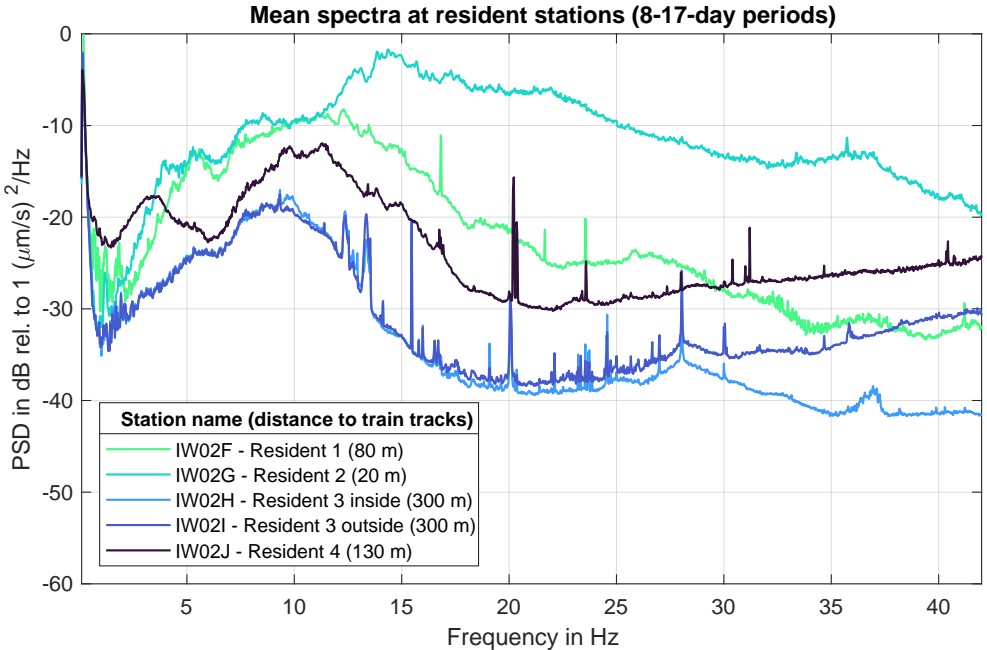

**Figure 8.** Mean spectra for 8 to 17 day long measurements at residents in the municipality of Kuchen. Amplitude and frequencies exited are closely related to the distance of the measurement location to the train tracks. Two instruments were installed at resident 3, where lower amplitudes are measured in the basement compared to the outside location for frequencies above 25 Hz.



### 3.3 Comparison to acoustic signals

Side-by-side measurements of ground motion and acoustic signals allow for a direct comparison of how WT emissions and immissions manifest in both types of data (as underground and air-borne noise). We conducted both types of measurements simultaneously at approximately 150 m distance to WT 1 of wind farm Tegelberg (ground motions - IW02B, acoustics - IMC_A1) and at the four resident sites in Kuchen (ground motions - IW02F-J, acoustic - IMC_B1 and IMC_B2). Compared to the ground motion data, where 100 Hz was used as the sampling rate, the acoustic data is sampled with 20 kHz. This allows

to record frequencies of up to 10 kHz, although for the assessment of low-frequency noise only frequencies of 1-200 Hz are considered (Gaßner et al., 2022).

Figure 9 shows time series and spectra up to 45 Hz for eight hours of recording time at the site near WT 1 for 2020-10-30 (compare also Figure 3 where a shorter section is displayed). We can observe more uniform amplitudes in the ground motion data, while in the pressure data a higher number of transient signals (amplitudes of more than 5 Pa) is visible. Both, ground

motion and acoustic, spectra contain clear signals proportional to the rotation rate at 32×, 46× and 64×BPF. For frequencies below 12 Hz in the ground motion data signals at constant frequencies (eigen modes) can be observed while in the acoustic data noticeable signals at 1-8×BPF are visible. Below 10 Hz signals related to the train traffic through Kuchen are also present in the ground motion data but not in the acoustic data.

At resident 1 in Kuchen we can observe WT related signals for a time window of almost no train traffic on 2020-10-26

from approximately 01:00 UTC to 03:00 UTC (Figure 10). In the data of all three instruments (ground motion outside the house, acoustic outside and inside) the signal at 32×BPF can be observed. In the acoustic data also signals at 1-8×BPF are visible, especially from 04:00 UTC to 08:00 UTC for the outside as well as inside measurements. In the respective ground motion data signals with frequencies at the eigen modes at 1.2 Hz and 3.6 Hz are present. The train related signals (starting at approximately 03:00 UTC) are of much higher amplitude compared to the background signals than in the acoustic data and

make an identification of WT related signals almost impossible.

The distribution of maximum amplitudes for signals with frequencies at 32×, 46× and 64×BPF found in the acoustic and ground motion data at the measurement site in 150 m distance to WT 1 is shown in Figure 11. In the acoustic data the mean of the maximum signal amplitudes decreases uniformly for each rotation rate, while in the ground motion data a different behavior can be observed, with a more significant decrease in maximum amplitudes for frequencies above 20 Hz. This behavior also

affects the ratio of the amplitudes, the so called coupling transfer coefficient $C_{AS}$ (e.g., Novoselov et al., 2020)

$$C_{AS} = \max(A_{seis})/\max(A_{acoustic}). \tag{1}$$

A similar ratio of $C_{AS} \sim 16.5$ can be observed for frequencies of 12.5 Hz, 18 Hz, and 41 Hz, while $C_{AS} \sim 10$ is found for 20 Hz and $C_{AS} \sim 3-5$ for 26 Hz and 29 Hz, respectively. These values indicate how the signals could propagate. While they are obviously emitted as acoustic waves, they are also induced as vibrations through the foundation into the ground. The

different amplitude ratios could indicate that additionally acoustic signals are coupled into the ground with different admittance for different frequencies (compare Novoselov et al., 2020).





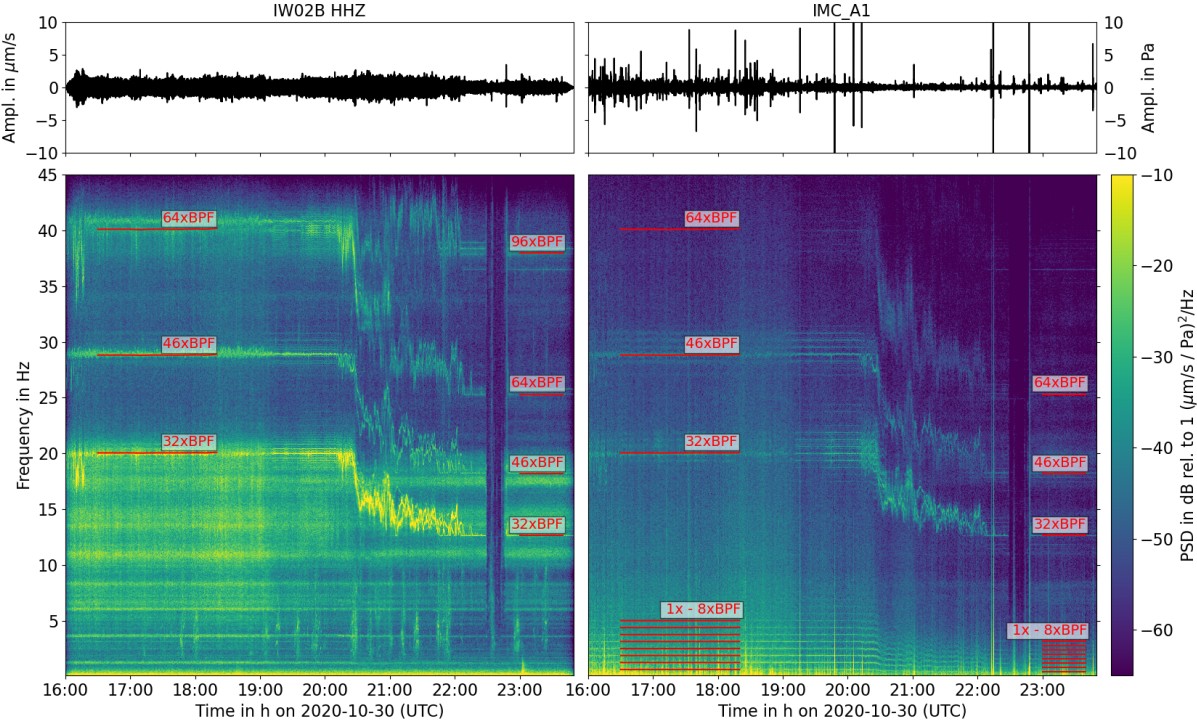

**Figure 9.** Time series and spectrograms for recording stations IW02B (left, vertical ground motion) and IMC_A1 (right, sound pressure) in 150 m distance to WT 1 of wind farm Tegelberg for 2020-10-30. A change in rotation rate can be observed at 20:30 UTC and a shutdown of approximately 10 min shortly before 23:00 UTC.

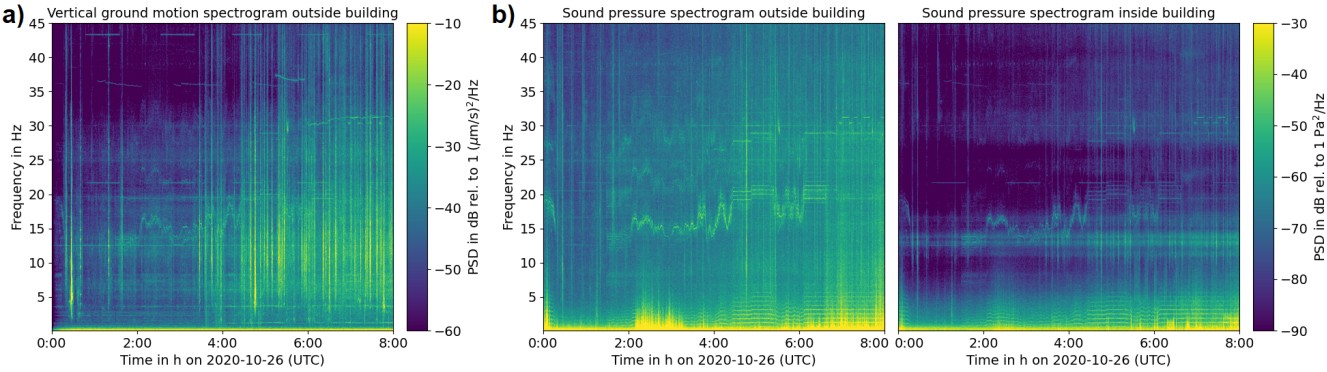

**Figure 10.** Spectrograms for sites a) IW02F (ground motion), b) IMC_B1 (acoustic indoors) and IMC_B2 (acoustic outdoors) at resident 1 for 2020-10-26 up to 45 Hz. Signals with frequencies 32×BPF can be observed in all three data sets. In the ground motion data train signals dominate before 01:00 UTC and after 03:00 UTC.



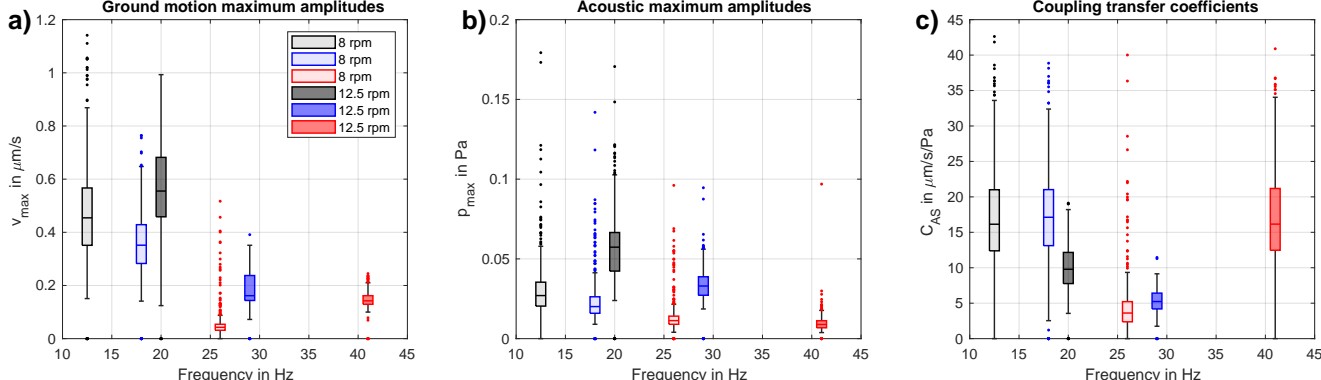

**Figure 11.** Distribution of maximum a) vertical ground motion and b) acoustic amplitudes for the frequencies marked in Figure 6 in 150 m distance to WT 1 of wind farm Tegelberg, with c) coupling transfer coefficients $C_{\mathrm{AS}}$ according to Novoselov et al. (2020).

## 4 Amplitude decay

An important goal of ground motion measurements in the vicinity of wind farms is the estimation of the amplitude-decay rate of the emitted signals (e.g., Stammler and Ceranna, 2016; Zieger and Ritter, 2018; Neuffer et al., 2019; Lerbs et al., 2020; Limberger et al., 2021; Neuffer et al., 2021). Amplitude-decay relationships can be used to predict the amplitude of WT-related ground motions in specific distances and, therefore, the influence on sensitive measuring equipment, e.g., seismological sensors. Typically the amplitudes $A$ of the respective PSDs (or rms-amplitudes, with $b_{\mathrm{rms}} = 0.5 \cdot b_{\mathrm{PSD}}$), that are measured at instruments along distances $r$, are fitted by a power-law decay with

$$A \sim 1/r^b. \tag{2}$$

This simple relationship summarizes the different damping relations (geometrical spreading, anelastic and scattering attenuation, and focusing effects) which can hardly be separated. Due to geometric spreading, a value of $b$=0.5 is expected for the decay of rms-amplitudes of surface waves which are most likely the main wave type emitted by WTs (Neuffer et al., 2021). In general, $b$-values typically increase for higher frequencies due to frequency-dependent anelastic attenuation and scattering. Therefore, resulting $b$-values are expected to differ significantly for different frequencies and geology, i.e., subsurface complexity and structure.

While varying geology is expected to be responsible for different $b$-values, results from other studies also differ considerably in the approach of fitting amplitude-decay curves. The distances considered range between up to 1 km (Neuffer et al., 2021) or at least 1 km to 8 km (Stammler and Ceranna, 2016). The chosen distance ranges naturally depend on the study design, available instruments and sites for measurements, and other noise sources (e.g., traffic), but are a significant factor in the estimation of decay curves. Zieger and Ritter (2018) have shown that $b$-values differ significantly when considering only distances of up to 100 m instead of 600 m. Furthermore, measurement times for PSD estimation last from hours (Zieger and Ritter, 2018)





to months (Limberger et al., 2021). Stammler and Ceranna (2016) use PSD/$N$ with $N$ the number of considered WTs, and take the harmonic mean of WT distances. This approach makes sense in their case, as they take data from stations in large distances of several kilometers to the WTs, where a wind farm can potentially be considered as a single signal source. In studies with measurements closer to the WTs mostly the distance to the nearest WT is considered, e.g., Zieger and Ritter (2018) and Limberger et al. (2021) use minimum distances (or only one WT) and furthermore the original PSD values without scaling.

Neuffer et al. (2019) use distances to the central WT of the investigated wind farm in their study and take relative PSD values ($\Delta$PSD) to exclude station specific noise effects, which is reasonable if constant noise sources, e.g., traffic, are present. For wind farm Tegelberg, where a railway line is close (compare Figure 1), this approach could mitigate the influence related to the distance to train tracks which is clearly visible in PSD spectra (compare Figure 10). For wind farm Lauterstein it might not be necessary to use $\Delta$PSD, as no other prominent constant noise sources can be identified in addition to the WTs. Furthermore, it is more difficult to find a quiet period due to the extent of the wind farm with many WTs and few stand-still periods. It also needs to be considered that the background PSD level for evaluation needs to be captured for a representative time, i.e., it should not include strong singular transient events. In this section we compare results for both approaches to estimate amplitude-decay relations.

In our evaluation we use six frequencies which comprise eigen modes (1.2 Hz, 3.6 Hz, 8.33 Hz, 11.25 Hz) and frequencies proportional to the BPF at full load (20 Hz, 29 Hz). For wind farm Lauterstein only frequencies up to 8.33 Hz are considered, as signals for higher frequencies are not as prominent in the data. We use frequency bands of $\pm 0.4$ Hz relative to the center frequencies. Additionally, for wind farm Tegelberg there are three time segments where only a single WT was running. This was either WT 1 (22 h and 12 h) or WT 3 (7 h). For these time segments, the direct distance of the recording stations to the respective WT can be used to determine $b$, leading to distances of up to 800 m. We compare $b$-values calculated directly from the PSD peaks (Figure 12, top row) to relative PSD values (compare Figure 12, bottom row). An improved fit of the PSD values to the amplitude decay curves can be observed for relative PSD values, which we attribute to the influence of the train traffic on seismic amplitudes. The resulting $b$-values can be found in Table 3.

For full operation of both wind farms we analyze both the maximum PSD values for two to three day periods and those values relative to values for times when the WTs were not running (two days at wind farm Tegelberg and 2.5 h at wind farm Lauterstein). PSD and $\Delta$PSD are plotted over the minimum distance of the respective instrument to the WTs (Figures 13 and 14). For wind farm Tegelberg these are distances of 150 m to 550 m and for wind farm Lauterstein 400 m to 1,900 m. At wind farm Tegelberg $b$-values of 0.5 at 1.2 Hz to $b$=3.6 at 20 Hz are found for single WT operation, while for full operation values range between $b$=0.1 at 1.2 Hz to $b$=4.6 at 20 Hz (compare Table 3, Figure 15). At wind farm Lauterstein we find $b$-values of 0.4 at 1.2 Hz to $b$=1.5 at 8.33 Hz (Figure 16). Comparing single WT to full wind farm operation, no clear trend can be observed between the respective $b$-values. For 20 Hz and 29 Hz $b$-values for full operation are increased, while $b$-values for lower frequencies are similar.

In Figure 17 $b$-values are compared graphically with symbol sizes proportional to the root-mean-square error (RMSE). The comparison shows that differences in $b$-values are significant for the different approaches, but RMSE values give no clear





indication which approach is to be preferred. We observe no significant difference for 1.2 Hz and 3.6 Hz, and a distinct benefit in $b$-value estimation mainly for 20 Hz and 29 Hz. Considering the results from ΔPSD values, the most significant discrepancy appears for 3.6 Hz where $b$-values for Tegelberg are $b$=1.7 (single WT operation) and $b$=1.9 (full wind farm operation), while

for wind farm Lauterstein a $b$-value of 0.3 is determined. Despite a strong scatter for PSD values at wind farm Lauterstein (Figure 14) $b$-values for 8.33 Hz at full operation agree well between Tegelberg and Lauterstein at $b$=1.5.

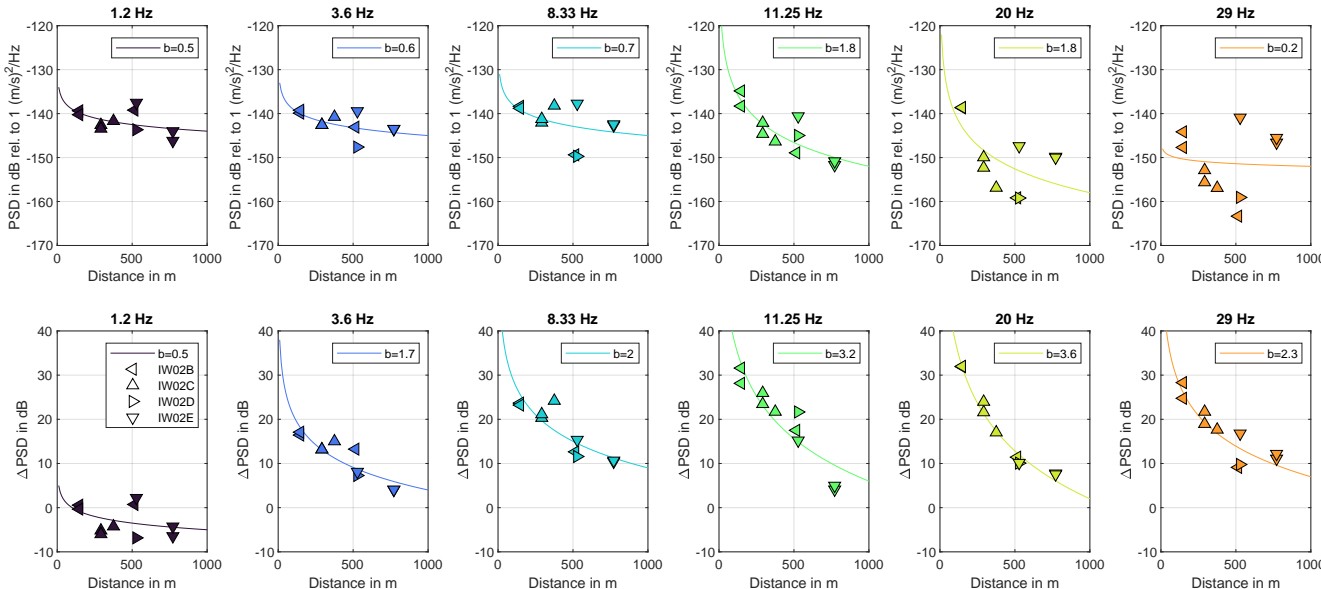

**Figure 12.** Amplitude decay estimation for three time intervals when only one WT was running at wind farm Tegelberg. Top: amplitude decay for PSD peaks. Bottom: amplitude decay for relative PSD values. The estimated $b$-values are graphically compared in Figure 17 and are listed in Table 3.

## 5   Discussion

The acceptance of wind energy by society is linked to the understanding of WT emissions which are of acoustic but also of ground motion nature. Such emissions can both be recorded in ground motion data due to mechanical coupling with the

ground (Figures 9 an 10). In this study we analyze ground motion emissions of two wind farms consisting of three and sixteen WTs, respectively (see Figure 1). At sites with 150 m to 1900 m distance to the nearest WT we observe signals below 12 Hz which are related to the eigen modes of the WT tower and blade rotation. Measurements at different wind farms with the same turbine type comfirm that the same frequencies are excited (Figure 5). The signals increase in amplitude with higher rotation rates (Figure 6) and can be observed over distances of several kilometers. Measured amplitudes of these eigen modes

(as well as other WT related signals, compare Figures 3 and 4) are at least one magnitude lower than the threshold for human





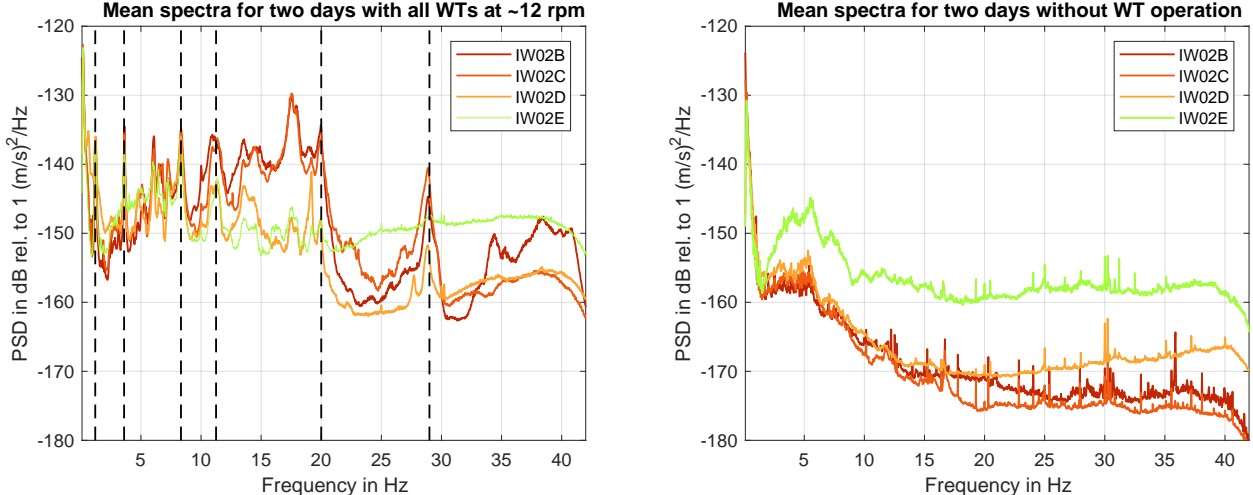

**Figure 13.** Spectra for amplitude decay estimation for wind farm Tegelberg with three WTs. Left: Mean spectra for sites IW02B-E for two days of full operation of the wind farm. Frequencies used for amplitude decay estimation are marked by black-dashed lines. Right: Mean spectra for two days without WT operation. Note the increased PSD levels at site IW02E which is closest to Kuchen and the railway line.

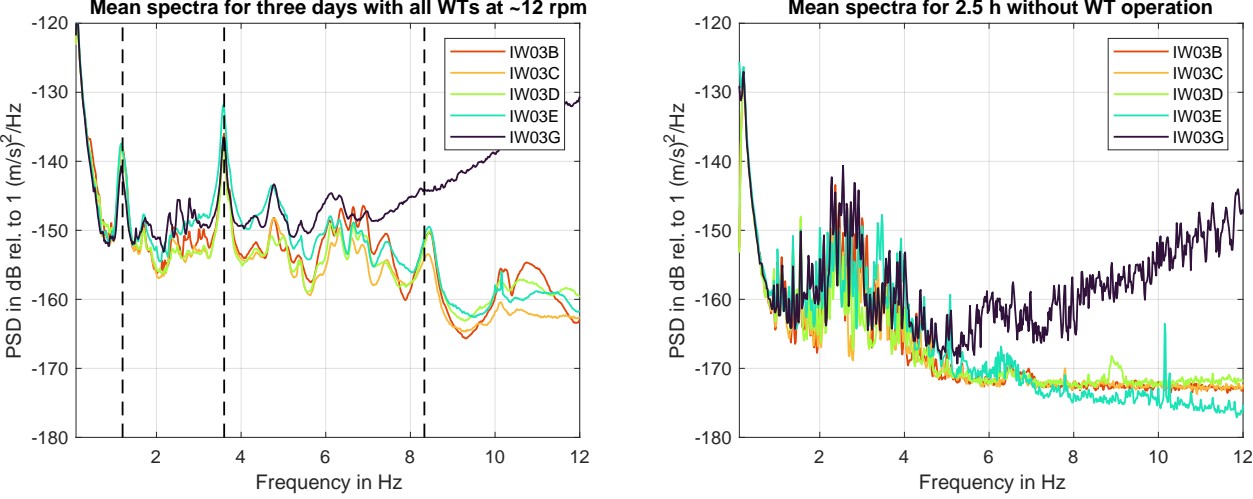

**Figure 14.** Spectra for amplitude decay estimation for wind farm Lauterstein with 16 WTs. Left: Mean spectra for instruments IW03B-E and IW03G for three days of full operation of the wind farm. Frequencies used for amplitude decay estimation are marked by black-dashed lines. Right: Mean spectra for 2.5 hours without WT operation. The increased PSD levels at site IW03G for frequencies above 5 Hz originate from a nearby gas heating.




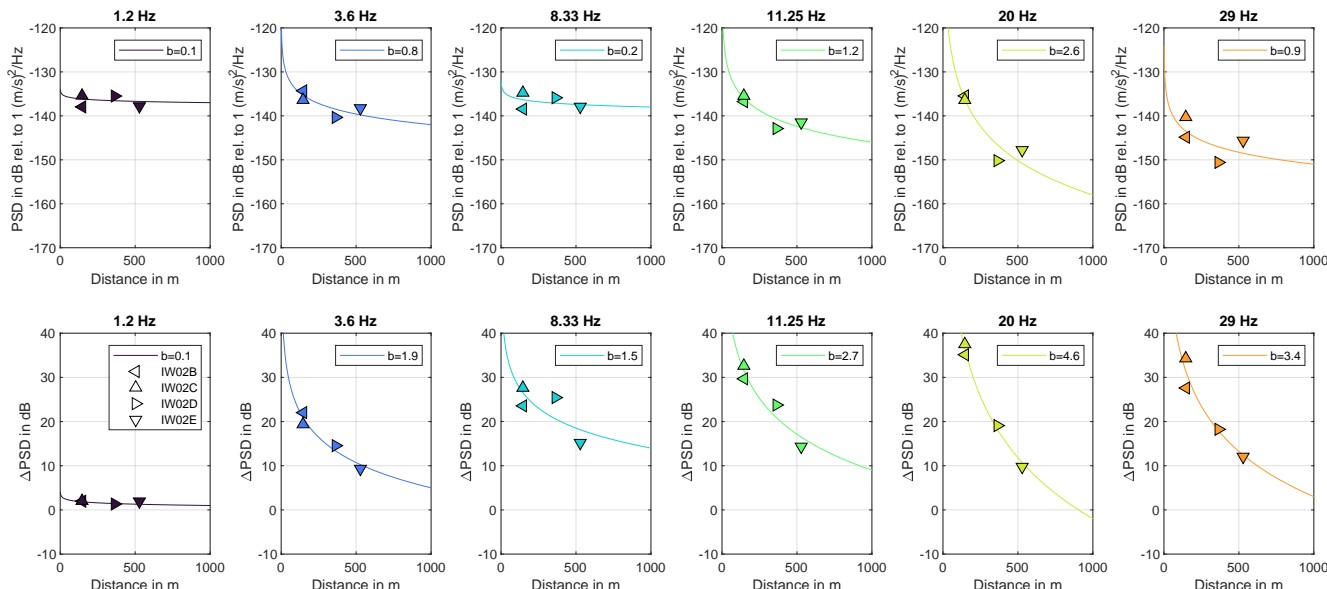

**Figure 15.** Amplitude decay estimation for full operation at wind farm Tegelberg. Top: amplitude decay for PSD peaks. Bottom: amplitude decay for relative PSD values. The estimated $b$-values are graphically compared in Figure 17 and are listed in Table 3.

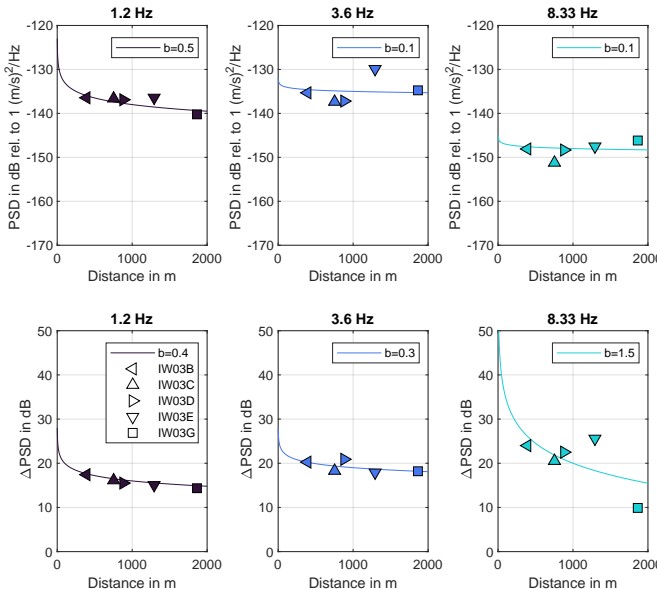

**Figure 16.** Amplitude decay estimation for full operation at wind farm Lauterstein. Top: amplitude decay for PSD peaks. Bottom: amplitude decay for relative PSD values. The estimated $b$-values are graphically compared in Figure 17 and are listed in Table 3.



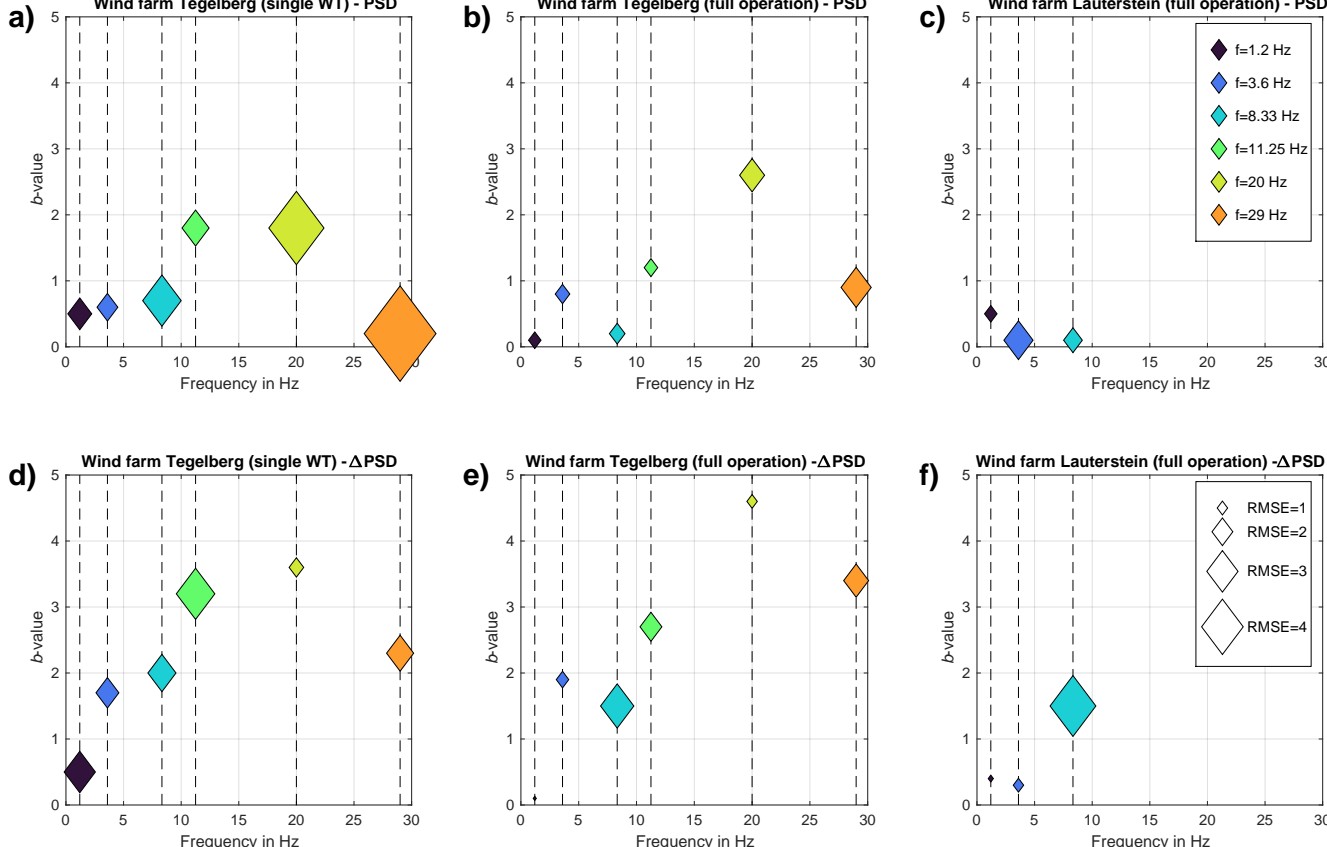

**Figure 17.** Graphical comparison of $b$-values for a) and d) single WT operation at wind farm Tegelberg, b) and e) full operation at wind farm Tegelberg, and c) and f) full operation at wind farm Lauterstein. a)-c) $b$-values for PSD peaks, d)-f) $b$-values for relative PSD. The size of the symbols corresponds to the RMSE of the measured PSD values compared to the theoretical ones from the fitted curves.

perception (∼100 $\mu$m/s). Therefore, local residents are not affected, but ground motions are relevant for sensitive equipment like seismological sensors.

Additionally, signals with frequencies proportional to the BPF above 12 Hz can be observed at distances of up to 1 km (e.g., Figure 3). These signals are also present in the acoustic data (Figure 9) and can be related to the WT generator speed

and potentially the gears. They are attenuated strongly and, therefore, only affect the direct vicinity of the wind farms but give indications to how WT signals are perceived by residents living near WTs who experience acoustic disturbances. Side-by-side acoustic and ground motion measurements also allow for the identification of other noise sources, especially train traffic in the municipality of Kuchen. We find that it has a much stronger influence on ground motion data than on the acoustic data, though. Furthermore, measured signals reflect short term changes in WT rotation rate which potentially capture the attention of



**Table 3.** Amplitude decay $b$-values, comparing the use of PSD and relative PSD values, found for full operation periods for wind farms Tegelberg (1.2 Hz to 29 Hz) and Lauterstein (1.2 Hz to 8.33 Hz). For wind farm Tegelberg additional results can be determined for single WT operation and are listed for comparison. Literature values are available from other studies in Germany for frequencies from 1.1 Hz to 7.6 Hz, focussing on ground motions due to eigenmode vibrations of the WTs.

| | 1.2 Hz | 3.6 Hz | 8.33 Hz | 11.25 Hz | 20 Hz | 29 Hz |
|---|---|---|---|---|---|---|
| Tegelberg (single WT, PSD) | 0.5 | 0.6 | 0.7 | 1.8 | 1.8 | 0.2 |
| Tegelberg (single WT, $\Delta$PSD) | 0.5 | 1.7 | 2.0 | 3.2 | 3.6 | 2.3 |
| Tegelberg (full wind farm, PSD) | 0.1 | 0.8 | 0.2 | 1.2 | 2.6 | 0.9 |
| Tegelberg (full wind farm, $\Delta$PSD) | 0.1 | 1.9 | 1.5 | 2.7 | 4.6 | 3.4 |
| Lauterstein (full wind farm, PSD) | 0.5 | 0.1 | 0.1 | | | |
| Lauterstein (full wind farm, $\Delta$PSD) | 0.4 | 0.3 | 1.5 | | | |
| | 1.1 Hz | 3-4.4 Hz | 5-5.5 Hz | 7.6 Hz | | |
| Stammler and Ceranna (2016) | | 2.7 | | | | |
| Zieger and Ritter (2018) | | 0.8 | 1.6 | | | |
| Neuffer et al. (2019) | | 2.4-2.9 | 4.4-4.9 | | | |
| Lerbs et al. (2020) | 0.7 | 1.1-1.3 | 2.3 | | | |
| Limberger et al. (2021) | 0.4 | 1.6 | | 3.9 | | |
| Neuffer et al. (2021) | 0.7 | 1.0 | | | | |

residents (especially at night and early morning times, compare Gaßner et al., 2022) and could be an explanation for reported annoyance, linked to changes in acoustic immissions.

The simultaneous measurement of ground motion and acoustic data allows a detailed analysis and signal amplitude comparison. Amplitude ratios, considering acoustic-to-ground motion coupling as proposed by Novoselov et al. (2020), indicate differing signal strength of acoustic and ground motion signals depending on frequency. In follow-up studies it would be interesting to look at even higher frequencies at co-located measurement sites. Signals with frequencies proportional to the BPF are mainly found in our measurements at wind farm Tegelberg while at wind farm Lauterstein we observe less dominant signals at higher frequencies due to larger distances (390 m to 1900 m) and less superposition of emitted waves because of the wider spatial distribution of WTs.

Amplitude decay relations are important to predict amplitudes at specific distances and are derived from measurements along profiles between the WTs and nearby settlements. We review approaches found in literature and compare resulting $b$-values for using the maximum PSD values at the studied frequencies or relative PSDs considering a background PSD level when no WTs are running. This is beneficial especially in the municipality of Kuchen where a railway line is close and strongly influences ground motion amplitudes at higher frequencies. As stated by Limberger et al. (2021), the latter approach yields increased $b$-values. Our results are in agreement with findings of Zieger and Ritter (2018), Lerbs et al. (2020), Limberger et al. (2021),

and Neuffer et al. (2021). To improve the amplitude decay estimation profiles should be measured to the North of wind farm Tegelberg, away from settlements and noise sources like traffic which would allow a better comparison with other studies.

At wind farm Lauterstein measurements for low frequency emissions (<12 Hz) should include larger distances as we expect that the wavefield will be highly complex due to the high number of WTs and a superposition of signals in the direct vicinity of the wind farm.

## 6 Conclusions

We conducted two 108- and 46-day-long measurement campaigns to study WT-related signals at two wind farms on the Swabian Alb and at resident sites in nearby communities. Observed signals comprise eigen modes and signals related to the BPF, while in the municipality of Kuchen train traffic is the most significant signal source in the ground motion data. Measured ground motion signals do not have perceptible amplitudes for humans (much smaller than 100 $\mu$m/s) even in the direct vicinity of the WTs.

Furthermore, our measurements were conducted in close proximity to acoustic measurements, allowing for a comparison of WT signals in both types of data. We can identify signals related to the WT generators and gears, which are potentially related to annoyance reported by residents and, therefore, provide a helpful data basis to evaluate noise reports. Our comparison of the ground motion and acoustic recordings find strong similarities above 12 Hz. Thus, recordings with seismometers and high sampling rates may be used in the future to monitor infrasound signals in the near-field of WTs.

At wind farm Tegelberg we observe strong signals with frequencies proportional to the BPF in the vicinity of the wind farm, which are less dominant at wind farm Lauterstein due to the wider station and WT distribution. Higher frequency signals are attenuated stronger with distance which manifests in the amplitude decay. We determine $b$-values for six and three frequencies at wind farm Tegelberg and Lauterstein, respectively. An approach considering relative PSD values is especially beneficial for data from wind farm Tegelberg where train traffic influences mainly the higher frequency amplitudes. Nevertheless, significant differences in $b$-values are observed for different approaches and discussions in the seismological community for a unified data analysis would improve the potential to predict ground motion amplitudes based on different geological settings.

Due to the complexity of wind farm Lauterstein with sixteen WTs the simulation of wave propagation, also considering the 3D topography, could give further indications on how to approach future amplitude-decay estimation.

*Code and data availability.* The code and data used in this research are currently restricted.

*Author contributions.* Laura Gaßner: Conceptualization, Data curation, Formal analysis, Investigation, Methodology, Software, Validation, Visualization, Writing - original draft. Joachim Ritter: Conceptualization, Funding acquisition, Project administration, Resources, Supervision, Validation, Writing - review & editing.





*Competing interests.* The authors declare that they have no known competing financial interests or personal relationships that could have
345  appeared to influence the work reported in this paper.

*Acknowledgements.* We thank the local authorities of the municipalities Kuchen and Degenfeld for their support. We also thank the Stadtwerke
Schwäbisch Hall and the KWA Contracting AG for providing access to WT 1 and the WT operating data at wind farm Tegelberg as well as
the wpd windmanager technik GmbH for access and data related to wind farm Lauterstein. We acknowledge the support of the local residents
who allowed the installation of instruments on their property and within their houses. Marie Gärtner and Leon Merkel assisted with the instal-
350  lation of the ground motion sensors. Esther Blumendeller (University of Stuttgart) provided the acoustic data used in this paper. Our study
benefits from the excellent cooperation between the Inter-Wind partners from MSH Medical School Hamburg, Martin-Luther-Universität
Halle, University of Stuttgart and the Centre for Solar Energy and Hydrogen Research Baden-Württemberg (ZSW) which is valuable for
successful interdisciplinary research. This study is supported by the Federal Ministry for Economic Affairs and Climate Action based on a
resolution of the German Bundestag (grant 03EE2023D).



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
