# Peer review of "Ground motion emissions due to wind turbines: observations, acoustic coupling, and attenuation relationships"

_EGUsphere, 2022_

## Author Response (AR1)

**RC1**

**The language needs to be improved. The structure of the abstract should be improved: both in form and language. In the first three lines there is too much repetition of the word emissions. It is not explained whether the words immission and emission are used interchangeably or whether they have any distinctive meaning.**

- We have changed the second sentence of the abstract to reduce the repetition of the word "emissions" and introduced the term "immission" here.

- It is not apparent to us where the language of our manuscript is suggested to be in need of improvement.

**There are problems of wording or use of words such as "measurements in the resident" that need to be improved before the manuscript is accepted.**

- No such wording as "measurements in the resident" is used in our manuscript.

**Sometimes it is said (compare figure XX) but it is not said what it is to be compared with.**

- We have exchanged "compare" for "see" in some references for clarity.

**RC2**

**In the following section ("Signals at place of immission"), the authors discuss signals which have been recorded in the village of Kuchen, in about 1 km distance from the closest wind farm. Obviously, some residents feel disturbed by the wind farm and one of the project goals lies on how residents experience the WT emissions. Unfortunately, it then turns out that the recordings in Kuchen cannot be evaluated, because here the wind turbine signals are completely covered by the vibrations of a nearby and heavily frequented railroad line. Therefore, I would shorten this section significantly and also remove Fig. 8 since, in my opinion, it does not contribute anything to the targeted questions.**

- We think it is a significant finding that the train signals are of such high amplitudes that the WT emissions are masked but the residents nevertheless claim to be disturbed rather by the WTs than the trains. Therefore, we would like to keep Figure 8 in the manuscript.

**From the amplitude ratio of infrasound and ground motion, the authors derive the so-called coupling transfer coefficient at different frequencies. The results are well documented by text and figures, however, I miss the discussion of the significance of this parameter, which is not very common in seismology. How is the mechanism of the transmission of airborne sound to ground motion, are there any insights on this? Or, which information can be derived from this parameter, what does this coupling factor depend on, e.g. properties of the subsurface? This section should be expanded accordingly.**

- We have added a passage to the manuscript to clarify this point (lines 220-225).

**One of the most important results of this manuscript is the presentation of the b-values describing the spatial decay of the seismic wind turbine emissions. There are already numerous publications on this topic, but the results vary quite strongly depending on e.g. the number of wind turbines, wind farm geometry, or geological conditions. I think each further experiment can help to bring systematics into the results and to better understand the emitted seismic wave field of a wind farm. Table 3 gives a nice compilation of published b-values. However I would also like to see a graphical representation, which enables a better and quick overview.**

- We added a graphical representation to the manuscript (Fig. 16) and simplified Table 3.

**To calculate the b-values, the authors use both, absolute PSD amplitudes and relative PSD values. The comparison shows that the relative PSD method results in somewhat more stable and more reliable b-values, particularly if the registration of wind turbine emissions is superimposed by transient noise. I would appreciate if this aspect would be discussed in more detail, as it could lead to unified measurement rules with which comparable b-values can be obtained in future.**

- We have added a passage to the manuscript to clarify this point (lines 292-294).

**In the discussion section there are some statements, which are not proofed by the presented results, concerning the range of the emitted wind turbine signals. At page 17, line 289 the authors write that the signals can be observed "over distances of several kilometers". However, no PSD recorded at several km distance from a wind turbine is shown in this manuscript.**

- We clarified this point in the manuscript (lines 306-308) and added a reference.

**Fig. 1: at WF Lauterstein there are 3 white WT symbols which means that these WTs were "not studied". - How is it possible to exclude them from the measurements?**

- The adjacent WF is of course responsible for part of the recorded signals, nevertheless, we have no operating data from those turbines. We rephrase to "no operating data available".

**page 12, line 182: ... The closest recording station (IW02F, 20m distance) ... - due to Fig.1 and Fig.8 the closest station is IW02G, IW02F is at 80m distance to the railroad track**

- This is correct and was changed accordingly.

**Fig.8: I would sort the legend entries by distance**

- Done.

**page 12, line 183: you should use just one unit for the PSD amplitudes throughout the manuscript (same in text and figures): either dB relative to 1 (m/s)\*\*2/Hz or dB relative to 1 (micrometer/s)\*\*2/Hz**

- The spectra have been changed to $(m/s)^2/Hz$, the spectrograms are kept with $(\mu m/s)^2/Hz$ to match the respective time series shown in units of $\mu m/s$.

**page 13, lines 192, 193: I think ground motions and acoustics were registered only at 2 resident sites simultaneously (IMC-B1 and IMC-B2)**

- The names IMC_B1 and IMC_B2 refer to the indoor and outdoor acoustic measurements, respectively. The names remained the same for each resident site. This is now explicitly explained in the text (lines 196-197).

**page 13, lines 199, 200; "Both, ground motion and acoustic, spectra contain clear signals …" - move the comma to "Both, ground motion and acoustic spectra, contain clear signals ..."**

- Done.

**page 13, lines 217, 218: please add the appropriate unit to the CAS values**

- Done.

**page 14, Fig.10, caption: "Signals with frequencies 32×BPF can be observed in all three data sets." - Please mark the addressed signals as you did in Fig. 9.**

- Done.

**page 17, line 286: "At sites with 150 m to 1900 m distance to the nearest WT …" - Obviously you are talking about WF Lauterstein. You should name it here.**

- Here we discuss both wind farms, closer stations (from 150 m) are at WF Tegelberg, stations up to distances of 1.9 km are at Lauterstein.

**page 18, line 287: "Measurements at different wind farms ...", the same as above. This is WF Tegelfeld.**

- Same as before, both wind farms are discussed.

**page 22, lines 329 330: "Thus, recordings with seismometers and high sampling rates may be used in the future to monitor infrasound signals in the near-field of WTs." - You cannot monitor infrasound with seismometers, can you?**

- Changed to "signals in the frequency range of infrasound".

**RC3**

**Abstract: Line 19: typo – … can be used to estimate …**

- Done.

**1. Introduction:**

**Line 25: 10 times the total height of the WT (to distinguish between hub height and total height)**

- Done.

**2. Measurements:**

**Line 66: For each campaign… - Where did you place the seismic sensors inside the WT tower? In the center of the foundation or at the outermost edge inside the tower? With regard to the comparison of amplitudes presented later in the study, this should be explained briefly to ensure comparability.**

- Sensors were placed at the outermost edge of the tower interior. A comment was added to clarify the exact location of the sensor (lines 68, 96, and 142-143).

**3. Ground motion signals:**

**Line 129: I agree that the train traffic signals are dominantly visible below 10 Hz, but they can also be detected at higher frequencies over the entire displayed frequency range (Fig 3d).**

- True, the text is changed accordingly.

**Line 140: Again, where did you place the sensor? (see comment above)**

- A comment was added to clarify the exact location of the sensor.

**Fig 5: This figure is a bit confusing and it is difficult to assess the shown results. I recommend using four different colors to represent the four sensor data and removing the distance arrow. Please add the distance information in the legend.**

- Done.

**3.1 Signals at place of emission:**

**Line 161 – 165: How exactly did you determine the maximum value of a 10-min segment? Did you simply identify the maximum absolute value of all values within a 10-min segment? How can you ensure that this value is due to WT vibrations and not to any other noise sources? I recommend to calculate the 95 % or maybe 99 % -percentile of all absolute values within each 10-min time window in order to get a more statistically representative value.**

- Yes, maximum amplitudes are taken per 10-min segment. It is a very quiet location and rarely transient signals are seen in the data which would not be connected to the WT operation. We checked that time windows are mostly free of these transient signals. By using approx. 360-550 time segments, we think trends seen are valid even if one or two time windows with signals not connected to WT operation would be included.

**3.2 Signals at place of immission:**

**Fig 8: Please sort the legend labels by distance between seismic station and WT.**

- Done.

**3.3 Comparison to acoustic signals:**

**Line 211 – 221: This part needs more details for clarification. Again, how did you calculate the max values for seismic and acoustic data (see comment above)? Does your method correspond to the method described in Novoselov et al. 2020 (they used the maximum of the envelopes of the respective signals)? What is the outcome of this analysis and what can we learn from these results?**

- We have added a passage to the manuscript to clarify this point (lines 220-231).

**Fig 11: I recommend to use different colors for each dataset of a given frequency and to include the multiples (32x, 46x and 64x) of the BPF in the legend.**

- Done.

**4. Amplitude decay:**

**Fig 13 and Fig 14: Please include the distance information for each station in the legend.**

- Done.

**Fig 16 and related text: I think it doesn't make sense to calculate amplitude decay curves and b-values from PSD data that are obviously not dominated by WT induced signals. How representative are the mean PSD spectra (Fig 14 right side) for periods without WT operation? I do not believe that this short period of only 2.5 h reliably represents the local noise level at the station sites. I recommend to remove Fig 16 and the corresponding text and to keep Fig 14 to describe why a decay value calculation for wind farm Lauterstein is not reasonable at this point.**

- We decided to move all graphics and text related to amplitude decay estimation at wind farm Lauterstein to the appendix.

**Most figures: I recommend to change the y-axis labels to a unified unit of "PSD in db rel. to 1 (m/s)²/Hz" for all figures showing PSD spectra.**

- Done.

**From a seismological point of view, it would also be nice if you could include the global models for high and low noise after Peterson (1993).**

- The NLNM/NHNM is not shown because it includes only frequencies of up to 10 Hz which is rarely suitable for the frequency range shown in this manuscript.

**Line numbers refer to the manuscript with markup!**

---

## Author Response (AR2)

Dear Susanne,

thank you a lot for taking over the editorship and taking time to read through the manuscript. Here are our answers to your review comments and suggestions.

**1. Reviewer 1 had the following suggestion: "What is novel in the paper is on the one hand the relation to the experiences of the residents, and on the other hand the analysis and comparison with acoustic signals. The authors should give much more emphasis and more space to these two aspects. The energy decay should be an adjunct to support this discussion and the conclusions about the problem with the residents."**

**This was not addressed in the rebuttal nor revision and i would ask that you incorporate this constructive suggestion.**

We moved the Lauterstein amplitude decay estimation to the appendix to reduce the section on amplitude decay estimation as suggested by reviewer 1. We missed to clearly point this out in the previous answer to the review.

**2. In the revision, most materials related to wind farm Lauterstein were moved to the Appendix. So even though sections 1, 2 and the beginning of section 3 discuss two wind farms (Lauterstein and Tegelberg), the results only discuss Tegelberg. I can see that there are reasons to do so, but these considerations need to be explained in the manuscript. This requires an adjustment of the Introduction and Measurements sections (where the expectation is raised that results and analyses of two wind farms will be presented) and a short addition to section 3 to explain why Lauterstein is not part of the main manuscript.**

We have now added text in sections 1-3 to better explain why the focus is on wind farm Tegelberg. We now explicitly describe this in the manuscript (lines 56-58 and lines 81-84).

**3. Tegelberg and the municipality Kuchen are close to a railway line and the measurements at Kuchen are disturbed by the trains. As this was to be expected, i assume that there were reasons to still select Tegelberg and Kuchen for this study (in the reply to reviewer 2 you hint at this). I suggest that the manuscript discusses the reasons why Tegelberg and Kuchen were still selected for this study, despite the railway line through the municipality.**

Our measurements at wind farm Tegelberg and in the village of Kuchen were directly co-located with acoustic measurements, and in general our focus is on this municipality because of the high number of annoyance reports. At wind farm Lauterstein there are almost no annoyance reports and no coincident acoustic measurements, This is now explicitly explained in the manuscript.

**Minor comments from marked draft:**

**- Suggest not to use abbreviations in abstract** – only the abbreviation for wind turbine (WT) is used and this is also used a lot in the main text, so we hope this is acceptable.

- A clarifying comment was added that there were 5 stations running permanently and 5 temporarily at the residents in Kuchen.

- Figure 2 was changed to a simpler, less ambiguous colorbar

- We are grateful for the rewording suggestions which have been adopted.